# Association between dental caries and *Helicobacter pylori* infection in Japanese adults: A cross-sectional study

**Komei Iwai**[1], **Tetsuji Azuma**[1], **Takatoshi Yonenaga**[1], **Kazutoshi Watanabe**[2], **Akihiro Obora**[2], **Fumiko Deguchi**[2], **Takao Kojima**[2], **Takaaki Tomofuji**[1] *

**1** Department of Community Oral Health, School of Dentistry, Asahi University, Mizuho, Gifu, Japan, **2** Asahi University Hospital, Gifu, Japan

* tomofu@dent.asahi-u.ac.jp

**Data Availability Statement:** Data cannot be shared publicly because of personal information protection. Data are available from the Department of Community Oral Health, School of Dentistry, Asahi University for researchers who meet the

## Abstract

*Helicobacter pylori* (*H. pylori*) is widely known as a cause of gastric disorders. Presence of *H. pylori* in dental pulp has been reported. Dental caries may influence the presence or absence of systemic *H. pylori* infection by serving as a source of *H. pylori*. In this cross-sectional study, we examined whether *H. pylori* infection in blood were associated with dental caries in Japanese adults. The participants were 752 individuals (513 males and 239 females, mean age 53.8 years) who underwent both *H. pylori* testing (*H. pylori* antibody test and pepsinogen test) and dental checkups at the Asahi University Hospital Human Health Center between April 2018 and March 2019. Those diagnosed as positive for *H. pylori* antibody test or positive for serum pepsinogen test as *H. pylori* test in the human health checkup were judged as those with *H. pylori* infection in the blood. In our study, 83 participants (11%) were determined to be infected with *H. pylori* in the blood. The proportion of those with decayed teeth was higher in participants with *H. pylori* infection in blood than in those without *H. pylori* infection in blood (p< 0.001). The logistic analysis showed that presence of *H. pylori* infection in blood was positively associated with those with decayed teeth (OR, 5.656; 95% CI, 3.374 to 9.479) after adjusting for age, gender, gastric disease, regular dental checkups, antibiotic medication history, and decayed teeth. Furthermore, the proportion of *H. pylori* infection in blood increased according to number of decayed teeth (p< 0.001). The results indicate that *H. pylori* infection in blood were associated with decayed teeth. Untreated dental caries may have an impact on systemic *H. pylori* infection.

## Introduction

*Helicobacter pylori* (*H. pylori*) is a Gram-negative spiral rod that causes gastric diseases such as chronic gastritis, gastric ulcer, duodenal ulcer, and gastric cancer [1–3]. *H. pylori* is an International Agency for Research on Cancer (IARC) Group 1 carcinogen [4] and the attributable fraction for gastric cancer is close to 90% [5]. The 2012 IARC report pointed out that more than half of the world's population is infected with *H. pylori* [4]. According to a 2012 report by Hirayama et al., the proportion of *H. pylori* infection among Japanese was about 28% in a

criteria for access to confidential data. Data requests can be sent to: ayumi-y@alice.asahi-u.ac.jp.

**Funding:** The authors received no specific funding for this work.

**Competing interests:** The authors have declared that no competing interests exist.

survey conducted between 2008 and 2011, and the proportion of infection decreased with decreasing age [6]. However, *H. pylori*-associated gastric cancer was the second most common cancer among men and the fourth most common among women in Japan in 2020, and it remains a disease that affects many people [7]. Therefore, prevention of *H. pylori* infection is a very important project in the field of public health in Japan.

In recent years, *H. pylori* has been detected in dental plaque, saliva, and pediatric dental pulp [8–10]. Our previous study also detected *H. pylori* in dental pulp and dental plaque of Japanese adults [11]. It is feasible that *H. pylori* infection in dental pulp was associated with systemic *H. pylori* infection.

Dental caries is a disease in which oral bacteria cause parenchymal defects in the tooth structure due to the acid produced from carbohydrates, and it is the most common cause of pulp infection [12]. It is reported that bacteria infected with dental pulp can migrate directly to blood circulation and cause atherosclerotic properties in blood vessels [13]. Like this, *H. pylori* infected with dental pulp due to dental caries may be detrimental to the risk of systemic *H. pylori* infection. However, there is very little literature on the relationship between dental caries and systemic *H. pylori* infection.

In our study, we hypothesized that dental caries might be associated with the risk of systemic *H. pylori* infection. The blood antibody tests are widely used to confirm the presence of *H. pylori* infection in the whole body [14]. Therefore, the aim of this cross-sectional study was to investigate the relationship between dental caries and *H. pylori* infection in blood in Japanese adults.

## Material and methods

### Participants

Participants were all those who had undergone both *H. pylori* testing (*H. pylori* antibody test and pepsinogen test) and dental checkups at Asahi University Hospital Human Health Center between April 2018 and March 2019, total of 782 people. Of these, participants with unknown *H. pylori* test results (27 participants) due to incompleteness at the time of the blood test and participants with unknown dental caries in the dental checkup (3 participants) were excluded from the analysis. As a result, 752 participants (513 males and 239 females, mean age 53.8 years) were included in the final analysis.

### Evaluation of the presence of *H. pylori* infection in the blood

Detection of *H. pylori* infection in the blood was done using *H. pylori* antibody test or pepsinogen test. In our study, those who tested positive for *H. pylori* antibody test or positive for serum pepsinogen test were determined to have *H. pylori* infection in their blood. It has been reported that *H. pylori* antibody test has a sensitivity of 97% and specificity of 95% [15], and pepsinogen test has a sensitivity of 93% and specificity of 91% [16]; these tests are currently widely used as tests for *H. pylori* infection [17].

### Oral examination

Four attending dentists checked the oral conditions, including number of decayed, missing teeth, and periodontal condition for each participant [18]. To evaluate periodontal condition, the periodontal probe (Hu-Friedy, USA) was used and the coded values of the Community Periodontal Index (CPI) were utilized [19]. The maximum value was classified into three code values (code 0 = periodontal pocket less than 3 mm, code 1 = periodontal pocket 4–5 mm, code 2 = periodontal pocket more than 6 mm), and codes 1 and 2 were evaluated as having

periodontal disease. Calibration was performed until the inter-rater agreement (kappa value) exceeded 0.8 with respect to the diagnostic criteria.

## Self-administered questionnaire

A self-administered questionnaire was used to investigate the participants' age, gender, smoking habits, drinking habits, medical history, sleep disorders, regular dental checkups, and antimicrobial medications. For smoking habits, those who smoked at least one cigarette per day were included (yes or no) [20]. For alcohol consumption, those who regularly drink alcohol at least once a week were included (yes or no) [21]. For regular dental checkups, those who regularly visit a dentist at least once six months were included (yes or no) [22].

## Statistical analysis

Significant differences in the characteristics of the presence or absence of *H. pylori* infection in blood were assessed using the chi-square test and Mann-Whitney U test. Univariate and multivariate stepwise logistic regression analyses were performed with the presence of *H. pylori* in blood infection as the dependent variable. The third category of variables related to the sample (age, gender) and the other variables related to *H. pylori* infection in blood (gastric disease, regular dental checkups, antibiotic medication history, and decayed teeth), which were adjusted for in these analyses. Differences in the prevalence of *H. pylori* infection in blood among different proportion of those with decayed teeth were assessed using the chi-square test; variables with p > 0.10 were excluded from the model, and variables with p < 0.05 were included in the model. All data were analyzed using a statistical analysis software (SPSS statistics version 27; IBM Japan, Tokyo, Japan). All p-values < 0.05 were considered statistically significant.

## Research ethics

Our study was approved by the Ethics Committee of Asahi University (No. 27010), and was performed in accordance with the Declaration of Helsinki. All residents who participated provided written informed consent. Our study is a cross-sectional study and follows the STROBE guidelines.

## Results

Table 1 shows the characteristics of our study participants with and without *H. pylori* infection in blood. In our study, 83 participants (11%) were positive for *H. pylori* infection in blood. The participants with *H. pylori* infection in blood were characterized by significantly higher proportion of those with decayed teeth (p = 0.001) compared to uninfected participants. The participants with *H. pylori* infection in blood were characterized by higher proportion of regular dental checkups (p = 0.077) and were also lower antibiotic medication history (yes; p = 0.096) compared to uninfected participants, but this was not significant. The results of univariate logistic regression analysis with *H. pylori* infection in blood as the dependent variable are shown in Table 2. The results showed that the presence of *H. pylori* in blood infection was significant associated with those with decayed teeth (OR, 4.929; 95% CI, 2.998 to 8.114).

Table 3 shows the adjusted odds ratios and 95% CI for *H. pylori* infection in blood according to the analyzed factors in participants. The participants with *H. pylori* infection in blood were significantly associated with those with decayed teeth (OR, 5.017; 95% CI, 3.031 to 8.305) after adjusting for age and gender. After additional adjustments for gastric disease, regular dental checkups, antibiotic medication history, and decayed teeth, the participants of *H. pylori*

**Table 1. Characteristics of the study participants with and without *H. pylori* infection in blood.**

| Factor | *H. pylori* infection in blood | | *p* value[*] |
|---|---|---|---|
| | **Absence (n = 669)** | **Presence (n = 83)** | |
| Age (years) | | | |
| -49 | 236 (35%) | 35 (42%) | 0.256 |
| 50–59 | 230 (34%) | 30 (36%) | |
| 60- | 203 (31%) | 28 (22%) | |
| Male [a] | 453 (68%) | 60 (72%) | 0.398 |
| Smoking habits [b] | 74 (11%) | 12 (15%) | 0.359 |
| Drinking habits [b] | 252 (38%) | 27 (33%) | 0.361 |
| Hypertension [b] | 36 (5%) | 2 (2%) | 0.244 |
| Diabetes [b] | 18 (3%) | 2 (2%) | 0.881 |
| Gastric disease [b] | 104 (16%) | 9 (11%) | 0.258 |
| Heart disease [b] | 43 (6%) | 3 (4%) | 0.313 |
| Sleep disorder [b] | 142 (21%) | 22 (27%) | 0.272 |
| Regular dental checkups [b] | 57 (9%) | 12 (15%) | 0.077 |
| Antibiotic medication history [b] | 92 (14%) | 6 (7%) | 0.096 |
| Periodontal pocket (mm) [b] | | | |
| -3 | 265 (40%) | 39 (47%) | 0.196 |
| 4- | 404 (60%) | 44 (53%) | |
| Gingival bleeding [b] | 340 (51%) | 47 (57%) | 0.318 |
| Number of present teeth [b] | | | |
| -20 | 19 (3%) | 1 (1%) | 0.644 |
| 21–23 | 28 (4%) | 4 (5%) | |
| 24- | 622 (93%) | 78 (94%) | |
| Decayed teeth [b] | 79 (12%) | 33 (40%) | < 0.001 |
| Missing teeth [b] | 268 (40%) | 41 (49%) | 0.103 |
| Filled teeth [b] | 654 (98%) | 82 (99%) | 0.537 |

[*] *p* < 0.05, using the Fishers exact test or the Mann-Whitney *U* test.

[a] Male (proportion of male)

[b] presence (proportion of presence).

infection in blood were significantly associated with those with decayed teeth (OR, 5.656; 95% CI, 3.374 to 9.479).

Table 4 shows difference in the proportion of *H. pylori* infection in blood according to number of decayed teeth. The proportion of *H. pylori* infection in blood among participants with one decayed tooth was 20% (13/66), among participants with two decayed teeth was 33% (10/30), and among participants with three or more decayed teeth was 63% (10/16), respectively. The proportion of *H. pylori* infection tended to increase with the number of decayed teeth (p < 0.001).

## Discussion

To the best of our knowledge, this was the first study to examine the association between dental caries and *H. pylori* infection in blood in Japanese adults. The results showed that the participants with *H. pylori* infection in blood had higher proportion of those with decayed teeth than those without *H. pylori* infection in blood. The logistic regression analyses also revealed that presence of *H. pylori* infection in blood was associated with those with decayed teeth after adjusting age, gender, gastric disease, regular dental checkups, antibiotic medication history,

**Table 2. Crude odds ratios and 95% CI for *H. pylori* infection in blood.**

| Factor | | Crude ORs | 95% Cl | *p* value |
|---|---|---|---|---|
| Age (years) | -49 | 1 | (reference) | 0.219 |
| | 50- | 0.747 | 0.470–1.188 | |
| Gender | Female | 1 | (reference) | 0.399 |
| | Male | 0.804 | 0.484–1.335 | |
| Smoking habits | No | 1 | (reference) | 0.361 |
| | Yes | 1.359 | 0.704–2.623 | |
| Drinking habits | No | 1 | (reference) | 0.361 |
| | Yes | 0.798 | 0.491–1.296 | |
| Hypertension | No | 1 | (reference) | 0.257 |
| | Yes | 0.434 | 0.103–1.837 | |
| Diabetes | No | 1 | (reference) | 0.881 |
| | Yes | 0.893 | 0.203–3.919 | |
| Gastric disease | No | 1 | (reference) | 0.261 |
| | Yes | 0.661 | 0.621–1.361 | |
| Heart disease | No | 1 | (reference) | 0.546 |
| | Yes | 0.546 | 0.166–1.800 | |
| Sleep disorder | No | 1 | (reference) | 0.273 |
| | Yes | 1.338 | 0.795–2.255 | |
| Regular dental checkups | No | 1 | (reference) | 0.081 |
| | Yes | 1.815 | 0.929–3.544 | |
| Antibiotic medication history | No | 1 | (reference) | 0.102 |
| | Yes | 0.489 | 0.207–1.154 | |
| Periodontal pocket (mm) | -3 | 1 | (reference) | 0.198 |
| | 4- | 0.740 | 0.468–1.170 | |
| Gingival bleeding | No | 1 | (reference) | 0.319 |
| | Yes | 1.263 | 0.798–2.001 | |
| Number of present teeth | -23 | 1 | (reference) | 0.735 |
| | 24- | 1.179 | 0.455–3.053 | |
| Decayed teeth | No | 1 | (reference) | < 0.001 |
| | Yes | 4.929 | 2.995–8.114 | |
| Missing teeth | No | 1 | (reference) | 0.104 |
| | Yes | 1.461 | 0.925–2.307 | |
| Filled teeth | No | 1 | (reference) | 0.543 |
| | Yes | 1.881 | 0.245–14.424 | |

Abbreviations: ORs, odds ratios; CI, confidence interval.

and decayed teeth. Furthermore, the proportion of *H. pylori* infection in blood increased according to the number of decayed teeth. These suggest that the risk of *H. pylori* infection in blood increases as the those with decayed teeth, and as the number of decayed teeth increases.

It has been reported that caries cavity is difficult to reach with a brush during oral cleaning, making it difficult to remove accumulated oral bacteria [23]. Therefore, caries cavity could serve as a reservoir for *H. pylori* because of self-cleaning is difficult to work, contributing to induce systemic *H. pylori* infection. Although further studies are needed, early treatment of decayed teeth may be beneficial to reduce the risk of systemic *H. pylori* infection. A previous study has reported that people who harbor severe dental caries have a higher detection rate of *H. pylori* in their saliva than those who do not [24]. The other study of deciduous teeth of 4–7

**Table 3. Adjusted odds ratios and 95% CI for *H. pylori* infection in blood.**

| Factor | | Adjusted ORs | 95% Cl | *p* value |
|---|---|---|---|---|
| **Model 1** | | | | |
| Age | -49 | 1 | (reference) | 0.177 |
| | 50- | 0.725 | 0.454–1.157 | |
| Gender | Female | 1 | (reference) | 0.318 |
| | Male | 0.770 | 0.462–1.285 | |
| Gastric disease | No | 1 | (reference) | 0.307 |
| | Yes | 0.683 | 0.328–1.421 | |
| Regular dental checkups | No | 1 | (reference) | 0.056 |
| | Yes | 1.939 | 0.982–3.830 | |
| Antibiotic medication history | No | 1 | (reference) | 0.121 |
| | Yes | 0.504 | 0.212–1.198 | |
| Decayed teeth | No | 1 | (reference) | < 0.001 |
| | Yes | 5.017 | 3.031–8.305 | |
| **Model 2** | | | | |
| Age | -49 | 1 | (reference) | 0.095 |
| | 50- | 0.656 | 0.401–1.076 | |
| Gender | Female | 1 | (reference) | 0.578 |
| | Male | 0.860 | 0.506–1.462 | |
| Gastric disease | No | 1 | (reference) | 0.267 |
| | Yes | 2.137 | 0.551–8.566 | |
| Regular dental checkups | No | 1 | (reference) | 0.414 |
| | Yes | 1.359 | 0.651–2.835 | |
| Antibiotic medication history | No | 1 | (reference) | 0.065 |
| | Yes | 0.226 | 0.046–1.097 | |
| Decayed teeth | No | 1 | (reference) | < 0.001 |
| | Yes | 5.656 | 3.374–9.479 | |

Abbreviations: ORs, odds ratios; CI, confidence interval.

Model 1: Adjustment for age and gender.

Model 2: Adjustment for age, gender, gastric disease, regular dental checkups, antibiotic medication history, and decayed teeth.

years old children also reported that subjects with a higher number of cavities had a higher proportion of *H. pylori* infection [25]. These observations are consistent with the present study which showed a significant association between dental caries and *H. pylori* infection.

In our study, the proportion of *H. pylori* infection in blood among all participants was 11%. This value was lower than previously reported the proportion of *H. pylori*-infected subjects in Japan [6, 26]. This may be due to the fact that the average age of the participants in our study was 53.8 years, which was lower than in past studies.

In our study, we find no association between Periodontal pocket and *H. pylori* infection in blood. This observation is agreement with previous studies, which reported that there was no

**Table 4. Differences in the proportion of *H. pylori* infection in blood according to different number of decayed teeth.**

| Factor | Number of decayed teeth | | | | *p* value |
|---|---|---|---|---|---|
| | 0 (n = 640) | 1 (n = 66) | 2 (n = 30) | 3- (n = 16) | |
| *H. pylori* infection in blood | 50 (8%) | 13 (20%) | 10 (33%) | 10 (63%) | < 0.001 |

Using the chi-square test.

significant correlation between *H. pylori* infection and severity of periodontitis [11, 27]. However, a previous study also has reported that a higher proportion of *H. pylori*-infected patients had periodontal disease compared to *H. pylori*-uninfected patients [28]. Thus, there is still no consensus on the relationship between *H. pylori* infection and periodontal condition. This can be attributed to several methodological differences, such as study design, sample population characteristics, and even regional differences [27].

In our study, there was no significant difference in the history of gastric diseases among those infected with *H. pylori* in their blood compared to those who were not infected. According to a report from a large cohort study of approximately 110,000 residents, those with the history of *H. pylori* infection had a significantly higher incidence of gastric disease than those without the history of gastric disease [29]. This may be due to the fact that the participants in our study are young in age. The preferred age for gastric cancer and gastric ulcer is 50 years or older, and the incidence increases over time [30]. We plan to continue to investigate the incidence of gastric diseases in *H. pylori* infection in blood over time.

In our study, participants with regular dental checkups were more likely to be infected with *H. pylori* than those without regular dental checkups. Since participants with decayed teeth in our study were more likely to have regular dental checkups than those without decayed teeth, there might be an indirect trend between regular dental checkups and the presence of *H. pylori*.

Our present study has some limitations. First, since this was a cross-sectional study, the timing of dental caries incidence was not confirmed. In the future, we should investigate the effects of period without dental caries treatment and dental caries severity on *H. pylori* infection in blood. Secondly, we did not evaluate dental caries severity. Since the size of caries cavity could have the effects on the relationship between dental caries and systemic *H. pylori* infection, the evaluation of dental caries severity (i.e., the International Caries Detection and Assessment System codes) may be important to improve the reliability of our observations. Finally, there is a possibility that false-positive participants in our antibody test were included. In the future, confirmation by urea breath test and gastric endoscopy will be considered for those who are positive for *H. pylori* infection in blood.

## Conclusions

The results of our study showed that *H. pylori* infection in blood was associated with decayed teeth in Japanese adults. Furthermore, people with a higher number of decayed teeth were more likely to have *H. pylori* infection in blood. Although further research is needed, recommending dental treatment for decayed teeth may have an impact on preventing systemic *H. pylori* infection.

## Acknowledgments

Our study was supported in part by Grant-in-Aid for Young Scientists (20K18809) from the Ministry of Education, Culture, Sports, Science and Technology of Japan.

## Author Contributions

**Data curation:** Takatoshi Yonenaga, Kazutoshi Watanabe, Akihiro Obora, Fumiko Deguchi, Takao Kojima.

**Funding acquisition:** Komei Iwai.

**Investigation:** Komei Iwai, Tetsuji Azuma.

**Methodology:** Komei Iwai.

**Project administration:** Takaaki Tomofuji.

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
