## [Decision Letter · Decision Letter 0]

8 Mar 2022

PONE-D-21-36232Association between dental caries and Helicobacter pylori infection in Japanese adults: A cross-sectional studyPLOS ONE

Dear Dr. Tomofuji,

Thank you for submitting your manuscript to PLOS ONE. After careful consideration, we feel that it has merit but does not fully meet PLOS ONE’s publication criteria as it currently stands. Therefore, we invite you to submit a revised version of the manuscript that addresses the points raised during the review process.

Please provide clarifications and explanations for the points raised by the reviewers. 

We look forward to receiving your revised manuscript.

Kind regards,

Iddya Karunasagar

Academic Editor

PLOS ONE

Journal Requirements:

"no competing interests"

 This information should be included in your cover letter; we will change the online submission form on your behalf

Additional Editor Comments (if provided):

The reviewers have identified some gaps in the study and asked for some clarifications. Please revise considering all reviewer comments.

Reviewers' comments:

Reviewer's Responses to Questions

**Comments to the Author**

1. Is the manuscript technically sound, and do the data support the conclusions?

Reviewer #1: Yes

Reviewer #2: Yes

2. Has the statistical analysis been performed appropriately and rigorously? 

Reviewer #1: Yes

Reviewer #2: Yes

3. Have the authors made all data underlying the findings in their manuscript fully available?

Reviewer #1: Yes

Reviewer #2: No

4. Is the manuscript presented in an intelligible fashion and written in standard English?

Reviewer #1: Yes

Reviewer #2: Yes

5. Review Comments to the Author

Reviewer #1: This is a good work. Please find my little remarks attached to the PDF. Please add to the discussion part a justification why patients who experience regular dental checks had more tendency to be infected with H pylori. You may also recommend the use of ICDAS caries scoring to study the severity of caries in relation to H pylori infection even though I understand that this was a retrospective study.

Reviewer #2: 1. Reduce your introduction

2. Please mention how you arrive at the sample size.

3. mention which sampling strategy you have adopted, whether universal or purposive

4. Please add the public health significance in conclusion section

5. Please explain the first sentence of Conclusion, need to check the sentence, grammar section.

6. It will be useful if any specific message targets towards practice of family physicians. What a family physician should check for and do to prevent dental caries

7. can you mention why significantly less number of females in comparison to male patients, any specific reason or habit responsible

6. PLOS authors have the option to publish the peer review history of their article (what does this mean?). If published, this will include your full peer review and any attached files.

Reviewer #1: **Yes: **Hisham Yehia El Batawi

Reviewer #2: **Yes: **Swayam Pragyan Parida

---

## [Author Response · Author response to Decision Letter 0]

8 Jun 2022

Response to Reviewers,

We greatly appreciate important comments by editor and reviewers gave us. We have introduced revisions to the text in accordance with reviewer’s comments, which we believe have considerably improved it. We have highlighted the changes to our manuscript using blue font.

PONE-D-21-36232

Title: Association between dental caries and Helicobacter pylori infection in Japanese adults: A cross-sectional study

Reviewers’ Comments:

Response to Reviewer 1

Please find my little remarks attached to the PDF; Do both tests have the same degree of sensitivity? Please elaborate on this.

Response: We thank the reviewer for this valuable recommendation. We have added the sentences regarding details on sensitivity and specificity of Helicobacter pylori (H. pylori) antibody test and pepsinogen test (lines 80-82).

Please find my little remarks attached to the PDF; In your study, you used the number of decayed and missing teeth which was a quantitative enumeration that does not reflect the severity of dental caries. You may hint to that in the discussion part.

Response: We thank the reviewer for this excellent suggestion. As you indicated, our study did not investigate dental caries severity. Since the size of caries cavity could have effects on the relationship between dental caries and systemic H. pylori infection, the evaluation of dental caries severity may be important to improve the reliability of our observations. This limitation has emphasized in the Discussion section (lines 198-202).

Please add to the discussion part a justification why patients who experience regular dental checks had more tendency to be infected with H. pylori.

Response: We thank the reviewer for this valuable recommendation. We have added the sentences (line 190-194).

You may also recommend the use of ICDAS caries scoring to study the severity of caries in relation to H. pylori infection even though I understand that this was a retrospective study.

Response: We thank the reviewer for this excellent suggestion. We did not use ICDAS caries scoring in the present study. However, we will use ICDAS caries scoring in our next research.

Response to Reviewer 2

1. Reduce your introduction.

Response: We thank the reviewer for this valuable recommendation. We have reduced introduction.

2. Please mention how you arrive at the sample size.

Response: Thank you for your valuable advice. Our study included all participants who had undergone both H. pylori testing (H. pylori antibody test and pepsinogen test) and dental checkups at Asahi University Hospital Human Health Center between April 2018 and March 2019 (lines 68-70). Therefore, we did not calculate the sample size. 

3. mention which sampling strategy you have adopted, whether universal or purposive.

Response: We thank the reviewer for this valuable recommendation. Our study employed a universal sampling technique.

4. Please add the public health significance in conclusion section.

Response: We thank the reviewer for this valuable recommendation. We have added the sentence (lines 208-210).

5. Please explain the first sentence of Conclusion, need to check the sentence, grammar section.

Response: We thank the reviewer for this valuable recommendation. We have revised the sentence (lines 206-208).

6. It will be useful if any specific message targets towards practice of family physicians. What a family physician should check for and do to prevent dental caries.

Response: We thank the reviewer for this valuable recommendation. Some people with dental caries do not undergo dental treatment. If dental caries is present and remains untreated, please encourage the patient to undergo dental treatment. We would also appreciate it if you could encourage them to encourage regular dental checkups to prevent dental caries.

7. Can you mention why significantly less number of females in comparison to male patients, any specific reason or habit responsible.

Response: We thank the reviewer for this valuable recommendation. In Japan, the proportion of women receiving health checkups tends to be lower than the proportion of men [Ministry of Health, Labour and Welfare (2019) National Survey of Living Standards. World Wide Website. Available: https://www.mhlw.go.jp/toukei/list/20-21kekka.html. Accessed 2022 Mar 18], which is like our study. The reason why the proportion of women receiving health checkups tends to be lower than the proportion of men may be due to their busy housework and child-rearing schedules, but it is not understood completely.

We answer in response to the reviewer's comments as noted above. We would like to express our sincere appreciation for your very useful suggestions. We look forward to working with you in the future.

---

## [Decision Letter · Decision Letter 1]

1 Jul 2022

Association between dental caries and Helicobacter pylori infection in Japanese adults: A cross-sectional study

PONE-D-21-36232R1

Dear Dr. Tomofuji,

We’re pleased to inform you that your manuscript has been judged scientifically suitable for publication and will be formally accepted for publication once it meets all outstanding technical requirements.

Kind regards,

Iddya Karunasagar

Academic Editor

PLOS ONE

Additional Editor Comments (optional):

All reviewer comments have been addressed.

Reviewers' comments:

Reviewer's Responses to Questions

**Comments to the Author**

1. If the authors have adequately addressed your comments raised in a previous round of review and you feel that this manuscript is now acceptable for publication, you may indicate that here to bypass the “Comments to the Author” section, enter your conflict of interest statement in the “Confidential to Editor” section, and submit your "Accept" recommendation.

Reviewer #1: All comments have been addressed

2. Is the manuscript technically sound, and do the data support the conclusions?

Reviewer #1: Yes

3. Has the statistical analysis been performed appropriately and rigorously? 

Reviewer #1: I Don't Know

4. Have the authors made all data underlying the findings in their manuscript fully available?

Reviewer #1: Yes

5. Is the manuscript presented in an intelligible fashion and written in standard English?

Reviewer #1: Yes

6. Review Comments to the Author

Reviewer #1: I think now the manuscript is ready for publication. I did not review the statistical analysis but it seems consistent.

7. PLOS authors have the option to publish the peer review history of their article (what does this mean?). If published, this will include your full peer review and any attached files.

Reviewer #1: **Yes: **Hisham Yehia ElBatawi

---

## [Editor Report · Acceptance letter]

6 Jul 2022

PONE-D-21-36232R1 

Association between dental caries and Helicobacter pylori infection in Japanese adults: A cross-sectional study 

Dear Dr. Tomofuji:

I'm pleased to inform you that your manuscript has been deemed suitable for publication in PLOS ONE. Congratulations! Your manuscript is now with our production department. 

Kind regards, 

on behalf of

Dr. Iddya Karunasagar 

Academic Editor

PLOS ONE